# Kraft Lignin/Tannin as a Potential Accelerator of Antioxidant and Antibacterial Properties in an Active Thermoplastic Polyester-Based Multifunctional Material

**DOI:** 10.3390/polym14081532

**Published:** 2022-04-09

**Authors:** Klementina Pušnik Črešnar, Alexandra Zamboulis, Dimitrios N. Bikiaris, Alexandra Aulova, Lidija Fras Zemljič

**Affiliations:** 1Faculty of Mechanical Engineering, University of Maribor, SI-2000 Maribor, Slovenia; klementina.pusnik@um.si; 2Laboratory of Polymer Chemistry and Technology, Department of Chemistry, Aristotle University of Thessaloniki, GR-541 24 Thessaloniki, Greece; azamboulis@gmail.com; 3Faculty of Mechanical Engineering, University of Ljubljana, Aškerčeva 6, SI-1000 Ljubljana, Slovenia; aulova@chalmers.se; 4Department of Industrial and Material Science, Chalmers Technical University, SE-41296 Gothenburg, Sweden

**Keywords:** poly (lactic acid), Kraft lignin, tannin, multifunctionality of PLA composites, surface mechanical properties, antioxidant/antibacterial activity

## Abstract

This research focuses on key priorities in the field of sustainable plastic composites that will lead to a reduction in CO_2_ pollution and support the EU’s goal of becoming carbon neutral by 2050. The main challenge is to develop high-performance polyphenol-reinforced thermoplastic composites, where the use of natural fillers replaces the usual chemical additives with non-toxic ones, not only to improve the final performance but also to increase the desired multifunctionalities (structural, antioxidant, and antibacterial). Therefore, poly (lactic acid) (PLA) composites based on Kraft lignin (KL) and tannin (TANN) were investigated. Two series of PLA composites, PLA-KL and PLA-TANN, which contained natural fillers (0.5%, 1.0%, and 2.5% (*w*/*w*)) were prepared by hot melt extrusion. The effects of KL and TANN on the PLA matrices were investigated, especially the surface physicochemical properties, mechanical properties, and antioxidant/antimicrobial activity. The surface physicochemical properties were evaluated by measuring the contact angle (CA), roughness, zeta potential, and nanoindentation. The results of the water contact angle showed that neither KL nor TANN caused a significant change in the wettability, but only a slight increase in the hydrophilicity of the PLA composites. The filler loading, the size of the particles with their available functional groups on the surfaces of the PLA composites, and the interaction between the filler and the PLA polymer depend on the roughness and zeta potential behavior of the PLA-KL and PLA-TANN composites and ultimately improve the surface mechanical properties. The antioxidant properties of the PLA-KL and PLA-TANN composites were determined using the DPPH (2,2′-diphenyl-1-picrylhydrazyl) test. The results show an efficient antioxidant behavior of all PLA-KL and PLA-TANN composites, which increases with the filler content. Finally, the KL- and PLA-based TANN have shown resistance to the Gram-negative bacteria, *E. coli*, but without a correlation trend between polyphenol filler content and structure.

## 1. Introduction

Sustainable and environmentally friendly production of polymer-based composites is increasingly promoted due to the accumulation of the most commonly used non-biodegradable highly durable conventional plastics from the petrochemical industry, leading to uncontrolled environmental pollution. According to the new vision for plastics in Europe under the European Strategy for a Circular Economy, polymer composites derived from natural resources are highly desirable to reduce CO_2_ emissions and the environmental footprint [1,2,3].

In this context, biodegradable poly (lactic acid) PLA, a thermoplastic polyester produced from renewable resources such as starch or sugarcane, has emerged as the frontrunner among the numerous polymer-based material composites that have been used for industrial applications as packaging materials or as a leading functional material in biotechnology and medicine (drug delivery, tissue engineering, 3D-printed devices) [4,5,6,7]. Compared to other conventional biopolymers, PLA has the following advantages: (i) environmental friendliness, (ii) biocompatibility, and (iii) processability and energy saving (20–25% less energy is consumed during PLA production compared to petrol-based polymers) [8,9,10].

However, some disadvantages limit its applicability: (i) low toughness, (ii) slow degradation rate, (iii) hydrophobicity, and (iv) chemical inertness due to the absence of reactive side chain groups. Therefore, with the incorporation of reinforcing agents into PLA in the field of nanomaterials or bulk material leading to a synergistic effect, the new technological development has overcome the limitations of PLA and in this way also improved the properties of PLA composites and introduced functionality [11,12,13,14,15,16]. Functional materials are designed to perform one or more necessary functions. Self-cleaning, conductive, magnetic, optical, and bioactive properties are of great interest for plastic composites, offering advantages over conventional plastics. In the packaging sector in particular, bioactivity in the form of antioxidant, antimicrobial, and barrier-forming properties is critical to extending the shelf life of food and ensuring its safety [17,18,19,20,21,22,23,24,25,26].

Considering the natural renewable reinforcements based on polyphenolic material, Kraft lignin (KL) and tannin (TANN) have great potential for the preparation of PLA composites to enhance functionality, especially toward bioactivity, whilst both phenolics act as free radical scavengers and thus as natural antioxidants that are UV resistant and bioactive [27,28]. In general, the incorporation of KL/TANN in polymer matrix exhibits several positive impacts as a compatibilizer, plasticizer, water repellent, flame retardant, and stabilizer; it also generates interest in the development of thermoplastic composites as adhesives and resins [29,30,31,32,33,34,35].

It is accepted that the antioxidant properties of KL/TANN correlate with the structural characteristics of polyphenols (*Mw*, molecular weight, polydispersity, functional groups) and their purity and depend on the size of the filler particles, the type of KL/TANN (grape tannin, pine bark tannin, mimosa tannin), and the loading mass [27,33,35,36,37]. In general, non-esterified hydroxyl groups cause high heterogeneity and polydispersity, resulting in lower antioxidant activity. In contrast, aromatic hydroxyl groups and high *Mw* exhibit high antioxidant activity. When a nanoscale filler is used, the enhanced antioxidant activity is attributed to the higher specific surface area. Grape tannins in 1% content in the polymer confer antioxidant properties and the effect increases steadily with TANN loading. In addition to grape TANN, pine bark and mimosa also play a role as effective short-term stabilizers [27,31].

In addition, it is hypothesized that the incorporation of KL/TANN in PLA could induce structural changes in PLA-KL/TANN composites on the one hand and modification of PLA surfaces on the other, which could not only improve the mechanical properties of PLA composites but also enhance hydrophilicity and roughness and further on confer antibacterial/antioxidant properties, which is a key strategy for packaging and functional bioengineering materials. A recent publication also elaborated that the crystalline morphology, structure, and interaction between the reinforcing agent and PLA affected the mechanical properties of the surface [37]. The amphiphilic behavior of KL improves filler-polymer compatibility, reduces polarity, and improves dispersion. At low or moderate wt% of the addition of KL, the mechanical performance is improved yet it depends on the chain length of the KL filler. On the other hand, several reports have found that the reinforcement of KL in PLA, Poly (butylene succinate) (PBS) and Poly (caprolactone) (PCl), and most aliphatic polyesters deteriorates the mechanical properties of lignin-based thermoplastic composites due to insufficient bonding between lignin and polymer [38,39,40]. To overcome this problem, chemical modifications have been carried out using coupling agents (e.g., esterification, acylation, and alkylation) and by organic solvent fractionation [39,40,41,42,43,44]. By incorporating TANN into PLA, the final properties also related to surface mechanical properties are associated with several problems: (i) the problem of incompatibility of hydrophilic TANN and polymer, leading to low adhesion of PLA-TANN composites, (ii) particle size control, and (iii) the mass fraction of TANN loading. The poor adhesion between hydrophilic TANN and hydrophobic polymer must be addressed, which leads to a severe decrease in mechanical properties. However, the addition of compatibilizers improves the interaction between TANN and polymer but poses some environmental and economic problems [27].

Although many studies have focused on the properties of polymer-based KL or TANN composites, no study has been published in the literature comparing the impact of KL/TANN natural additives on PLA-KL and PLA-TANN composite materials, without the addition of environmentally problematic toxic additives according to the sustainable attractive material from the point of view of safety and economy. In addition, there is the question of how admixture might affect the final synergistic end performance, including surface properties and mechanical and antibacterial/antioxidant activity. Surface properties are particularly important, because most plastics, especially those intended for packaging and biomaterials, work on the surface with contact media (food, tissue, etc.,). Therefore, taking into consideration the multifunctionality of PLA-KL and TANN composites, the objective of this work is primarily focused on comparing the overall performance of PLA composites processed by the hot melt extrusion method, improving the surface mechanical properties, and providing the antioxidant/antimicrobial activity. The dependence on two different polyphenols, KL and TANN, on the functional surface properties was investigated, whilst those PLA-based renewable thermoplastic composites filled with KL and TANN were already studied in terms of crystallization phenomena, interfacial interactions between PLA and KL, TANN, molecular mobility, and thermal degradation in our previous work [45,46].

In general, the surface mechanical properties of PLA-KL and PLA-TANN composite films were correlated to roughness as well as the contact angle measurement using goniometry, surpass zeta potential, and nanoindentation. Finally, the antioxidant/antimicrobial activity of the developed composites was also estimated using standard methods (DPPH• (2,2′-diphenyl-1-picrylhydrazyl), DSM 1576) [34,47]. Reinforcement with two different polyphenols, KL and TANN, resulted in promising multifunctionality of the material, allowing it to be used for both packaging and biomedical applications. However, the antioxidant reinforcing agents not only endow the composites with antioxidant activity but also improve their stability and prolong their shelf life. On the other hand, the additional antimicrobial activity of PLA composites provides a high quality microbial inhibiting material required for many advanced applications.

## 2. Materials and Methods

### 2.1. Materials

The polymer used in this study as the matrix was neat PLA, poly (L-lactic acid, with 96% of L and 4% of D isomer) and was provided by Plastika Kritis S.A. (Iraklion, Greece) with molecular weight Mw~75 kg/mol and intrinsic viscosity η = 1.24 dL/g. Kraft lignin (KL) and tannin (TANN) samples were acquired from Sigma-Aldrich. The physiochemical properties of KL and TANN were described in our previously published work [45,46].

#### PLA Composites Processing

KL and TANN were incorporated in different percentages (0.5, 1.0 and 2.5 wt%) into the PLA matrix. To remove moisture, PLA granulates, KL, and TANN were left in a vacuum oven overnight at 110 °C. PLA-based KL and TANN composites (PLA-KL, PLA-TANN) were prepared by melt mixing. The dried materials were prepared in a corotating twin screw melt mixer with 30 rpm at 195 °C for 10 min. The thin films (330–500 μm) of PLA-KL and PLA-TANN were obtained with thermal pressing at 180 °C with a Paul-Otto Weber (Paul-Otto Weber GmbH, Remshalden, Germany) thermal press and followed by a sudden cooling.

A complete overview of the prepared PLA-KL and PLA-TANN composites is presented in Table 1 with names for each composite mixture.

### 2.2. Methods

#### 2.2.1. PLA Composites Based on KL/TANN Characterization

##### Surface Mechanical Properties of PLA-KL and PLA-TANN Composites

To study the surface mechanical properties of PLA-KL and PLA-TANN composites, the contact angle measurements of water was followed. In total, the contact angle of all seven PLA composites (neat PLA and PLA-KL and PLA-TANN) with three different KL and TANN contents were measured using a goniometer from DataPhysics (Filderstadt, Germany). For this purpose, 3 μL of water was dropped onto the composite surfaces and an average of at least three liquid droplets per surface were calculated. In this way, the static contact angles (SCA) were measured at room temperature. The water contact angle determination of KL and TANN was also measured in the form of a pellet (Figure 1). The KL and TANN was dried overnight at 100 °C in a vacuum oven. The dried KL and TANN was ground into fine powder and a small amount of the KL and TANN powder was added to the collar of a Perkin Elmer Hydraulische Presse. The KL and TANN powder was pressed for 2 min to form a pellet. Contact angle measurements were performed using ultrapure water (Millipore, Burlington, MA, USA).

For the ***surface zeta potential as charge indication*** analysis with SurPASS-3 (Anton Paar GmbH, Graz, Austria), two sample pieces of approximately 10 mm × 10 mm were cut for the streaming potential measurement. The sample pieces were fixed on the sample holders of the Adjustable Gap Cell (with a cross-section of 20 mm × 10 mm) using double-sided adhesive tape. The sample pieces were aligned opposite each other such that the maximum surface area of the samples overlapped. The permeability index of the sample, by rotating the micrometer, was set to around 100. A 1 mM KCl electrolyte solution was used and the pH was automatically adjusted with 0.05 M NaOH and 0.05 M HCl. The pH dependence of the zeta potential was determined in the range of pH 2–10. A pressure gradient of 200–600 mbar was applied to generate the streaming potential. The distance between the sample pieces was adjusted to 105 ± 5 µm. Streaming potential measurements were performed using an aqueous 0.001 mol/L KCl solution as the background electrolyte. The pH dependence of the surface zeta potential was determined by adjusting an initial pH 10, using 0.05 mol/L KOH as reported elsewhere [48].

The ***surface roughness*** of PLA, PLA-KL, and PLA-TANN composites was measured using scratch testing with topographical compensation available for G200 Nanoindenter (Agilent, Santa Clara, CA, USA) that was already described in the literature [37]. Roughness values in nanometers have been extracted from the 500 µm long surface scan with an applied small force of 20 µN. Profiling velocity was set to 10 µm/s for every 10 scratches per material, which were located at 200 µm from each other. 

***Continuous Stiffness Measurement****is a* nanoindentation characterization method developed by Oliver and Pharr [49], which utilizes the dynamic loading of the sample material. CSM tests were performed using Nanoindenter G200 equipped with XP head according to protocol established in [37] and testing parameters are presented in Table 2. 

##### Antioxidant Properties of PLA-KL and PLA-TANN

The radical scavenging activity of neat PLA, PLA-KL, and PLA-TANN composites was evaluated using DPPH• (2,2′-diphenyl-1-picrylhydrazyl) (Sigma-Aldrich, France). The method of antiradical activities is based on the reduction of the DPPH• radical, which is analyzed spectrophotometrically at a wavelength of 517 nm (Spectrophotometer (UV-VIS) UV-1800 Shimadzu) as it was clearly introduced in our previously published paper [48]. The antioxidant capacity can be determined by the decrease of absorption at wavelength 517 nm. DPPH• radical can be reduced in the antioxidant (AO) presence, with the consequent decolorization from purple to yellow. DPPH solution was prepared in methanol (8.1 × 10^−5^ mol/L) at ambient temperature [27,50]. 

Briefly, all the samples neat PLA, PLA-KL, and PLA-TANN composite samples disks (1 cm × 1 cm) were directly immersed into 3 mL of the methanol DPPH• solution. The antiradical activities of the neat PLA, PLA-KL, and PLA-TANN films were determined straightaway after 1 h, 6 h, 12 h, and 24 h. The radical scavenging activity was calculated as per the following expression: Inhibition = (A_control_ − A_sample_)/A_control_ × 100%,
where A_Control_ is the absorbance measured at the starting concentration of DPPH• and A_Sample_ is the absorbance of the remaining concentration of DPPH• in the presence of PLA-KL and PLA-TANN composite polymers. The reference neat PLA material was also tested.

##### Antibacterial Properties of PLA-KL/PLA-TANN Composites

The microbiological analysis of neat PLA, PLA-KL, and PLA-TANN composites against the bacteria *Escherichia coli* (DSM 1576) as standard Gram-negative bacteria was performed according to the internal protocols standardized for plastic surfaces by the Department of Microbiological Research, Center for Medical Microbiology of the National Laboratory for Health, Environment and Food in Maribor, d.i.e., No. P96 Biofilm production on various materials P90 (ISO22196) reported elsewhere [51,52]. In brief, the neat PLA, PLA-KL, and PLA-TANN composite films of size 10 mm × 10 mm (approximately, 0.12 g of PLA composite material) were exposed to the standardized medium inoculated with *Escherichia coli* and set at 0.5 on the McFarland scale at three different incubation times: 1 h, 12 h, and 24 h. Then, after inoculation of the neat PLA, PLA-KL, and PLA-TANN film composites, the viable bacteria were evaluated by the pour plate method (plate counting agar was used). The effect of incorporation of polyphenolic filler loading into the PLA matrix composites was calculated as a reduction of bacterial growth and counting of bacterial number after incubation of neat PLA polymer compared with PLA-KL and PLA-TANN composite films.

## 3. Results and Discussion

### 3.1. Surface Mechanical Properties of PLA-KL and PLA-TANN Composites

The addition of polyphenolic fillers in PLA polymer such as KL and TANN leads on the one hand to changes, especially for packaging/biomedical applications, such as thermal behavior and degradation, semi-crystalline morphology, interfacial phenomena, and molecular mobility as published by our team [45,46], and on the other hand to improved functional properties with enhanced mechanical properties and antibacterial and antioxidant behavior of PLA-KL and PLA-TANN composites as shown here. To explore the distribution of KL and TANN in PLA polymer that causes the changes in PLA-KL and PLA-TANN surface behavior, the contact angle measurements, roughness measurements, zeta potential, and nanoindentation were investigated in more detail.

Figure 2a,b first shows the images of PLA-KL and PLA-TANN films with a homogeneous distribution of KL and TANN in the polymer material. Due to the native darker color of KL and TANN, the PLA composite material is darker, which could be explained by a suitable mixture between polyphenolic filler and PLA. 

The water contact angle measurement was applied to investigate the hydrophilicity/hydrophobicity of PLA-KL and PLA-TANN composites. Due to the low hydrophilicity of neat PLA (neat PLA showed contact angle values of 86.5 ± 2.5°), the incorporation of KL and TANN with predominant hydrophilic character into PLA could accelerate it [53,54,55,56]. The wettability of water with the surface of PLA-KL and PLA-TANN composite material is shown in Figure 2c. Neither KL nor TANN cause a significant trend in wettability. KL with the main phenolic and non-polar reactive groups (phenolic derivatives and aromatic hydrocarbons were identified) analyzed by Pyrolysis–gas chromatography/mass spectrometry (Py–GC/MS) and published in the previous study could enhance either the hydrophilic or the hydrophobic character of PLA-KL composites [45]. However, in our case the hydrophilic character was determined (CA of PLA-0.5KL = 83.2°, PLA-1.0KL = 87.8°, PLA-2.5KL = 79.2°). In contrast, TANN with polar phenolic groups was found to exhibit enhanced hydrophilicity. In general, all PLA composites with polyphenolic filler showed a change in contact angle compared with neat PLA, with a slight increase in hydrophilicity at the lowest KL and TANN concentration. The 0.5 wt.% addition of KL and TANN decreased the contact angle of PLA-KL by 5%. On the other side, the contact angle of PLA-0.5TANN composites was 17% lower than neat PLA, resulting in a higher hydrophilic nature of PLA composites. By increasing the KL concentration (1.0 wt.%) the water contact angle does not significantly change compared with neat PLA, but the 1% of TANN in PLA leads to an improved contact angle at 10%. Finally, further addition of TANN (i.e., 2.5%) increases the contact angle of PLA-TANN composites by 5%; in contrast, the PLA-KL composites CA decreased. 

The wettability of PLA-KL and PLA-TANN composites may be caused by the incorporation of KL and TANN and the interaction between neat PLA and KL/TANN filler and their available functional groups on the surfaces of PLA composites, which has already been explained in the literature [46]. Moreover, in the case of PLA-TANN, the hydrophilicity of TANN is more pronounced compared with KL (Figure 2c), which could be explained by the greater preferential interaction between the polar polyphenolic fillers of TANN than KL that exceeded an amphiphilic structure. Secondly, the wettability could be related to the differences in particle size of TANN and KL that provide available functional groups needed for interaction with PLA and a better distribution in PLA. Previous results of scanning electron microscopy (SEM) analysis determined the size of KL particles (600 nm) in comparison with TANN (450 nm). Thus, the results are consistent with those previously published [46] for PLA-polyphenol composites, but also differ from ours in terms of the origin and properties of KL and TANN and the processing of the final composite, which resulted in more hydrophilic composites (lower contact angle) compared with neat PLA [33,35,40,50,57].

The measurement of water contact angle has further been related to **surface roughness**, polymer, and filler chemistry. The average roughness of PLA, PLA-KL, and PLA-TANN composites is represented in Figure 3. Apparently, all samples, PLA, PLA-KL, and PLA-TANN, have a rough surface; however, the difference of material properties is still statistically significant according to ANOVA analysis. With the incorporation of KL and TANN, the roughness of the PLA composites changed and decreased with increasing polyphenol addition; more specifically, the PLA-TANN composites showed higher roughness than neat PLA and with increasing TANN loading the roughness of PLA-TANN decreased. In contrast, the roughness of PLA-KL showed approximately the same trend and did not change significantly. The smaller particles size of TANN and available functional groups at 0.5 wt.% of content of addition in PLA led to an increased interaction with PLA and, eventually, a more visible effect on the surface of the PLA-TANN composites. On the other hand, non-prominent interaction between KL and PLA neither in the small nor in the high concentration leading to agglomeration of KL in PLA does not much affect the roughness of PLA-KL film.

The roughness measurements of PLA-KL and PLA-TANN composites explain approximately the same trend of distribution of the polyphenolic filler in PLA polymer composites of metal oxide-based PLA composites in the PLA surface region and its roughness behavior [37]. This could be due to the incorporation and interaction of KL and TANN, which are stronger or more pronounced with decreasing loading of polyphenolic filler and lead to a higher availability in the inner region of PLA-KL and PLA-TANN composites, causing the changes in the surface region. 

In general, the trend of change in the water contact angle of PLA-KL and PLA-TANN composites agrees well with that of surface roughness, indicating the distribution of KL and TANN in PLA composites and varies in different deep layers of PLA-KL and PLA-TANN composites (Figure 4). From the correlation plot between CA and roughness (Figure 4), the correlation coefficient for PLA-TANN is high (0.973), indicating higher linear strength, while the lowest correlation coefficient is calculated for PLA-KL (0.715), which is still considered strong.

Surface charge expresses the interaction between PLA composites in aqueous solution and solid surfaces and is a key factor in explaining the differences between PLA composites before and after incorporation of the polyphenolic fillers. Therefore, the surface properties of PLA-based polyphenolic filler as varying the filler types and filler content were followed by zeta potential measurements. Figure 5a,b presents the zeta potential values of the neat PLA, PLA-KL, and PLA-TANN composites with addition of polyphenolic filler as a function of pH ranging from 2 to 12. The isoelectric point and the value of zeta potential are indicators of the functional groups present on the surface of the PLA composites. The decrease of the isoelectric point indicates the addition of acidic groups, while the increase indicates the addition of basic groups. The isoelectric point (IEP) (Figure 5a,b) was observed at pH = 3 for neat PLA; it is in accordance with the literature and it remains almost the same with KL and TANN incorporation in PLA (PLA-0.5KL = 2.90, PLA-1.0KL = 3.05, PLA-2.5KL = 3.21, PLA-0.5TANN = 3.11, PLA-1.0TANN = 2.75, PLA-2.5KL = 2.82) [58,59]. The negligible change in the IEP value of PLA-KL and PLA-TANN composites could be related to the PLA polymer and polyphenolic structure. Both PLA and the KL and TANN structures consist of COOH and OH groups. The interfacial interaction of PLA with KL and TANN, namely by the carbonyl group reported in our previous work, does not affect the charges and consequently yields a constant IEP value [48].

The total zeta potential of PLA-KL and PLA-TANN composites with high negative zeta potential at high pH towards more positive values was also determined in Figure 5. All PLA-KL and PLA-TANN above pH = 3 composites accelerate the negative zeta potential. The effects of the addition of KL and TANN in PLA on the surface and interfacial charge of PLA-KL and PLA-TANN are described by the negative zeta potential at pH = 9 (Figure 5a,b). It is clear that the zeta potential at pH = 9 plateau level follows a continuous trend related to the KL and TANN loading and distribution in the PLA matrix.

Due to the incorporation of KL and TANN into PLA, more charges are induced on the surface of PLA-KL and PLA-TANN, resulting in increased charge density, as shown by the increase of negative zeta potential plateau, but with decreasing KL and TANN loading. With increasing KL and TANN addition, the zeta potential at pH = 9 becomes more negative than neat PLA. The polar groups are introduced onto the surfaces of the composite, leading to an increase in PLA-KL and PLA-TANN surface hydrophilicity; also evidenced by the contact angle measurements. The zeta potential method correlates well with the measurement of the contact angle, which was shown in Figure 6 (the correlation coefficient for PLA-TANN is 0.91; for PLA-KL it was calculated at 0.79). In conclusion, the zeta potential measurements of all PLA-KL and PLA-TANN express anionic charge in almost the entire pH range, indicating a high affinity for adhesion/adsorption of cationic substances and repulsion of anionic substances, which may ultimately contribute to the antibacterial properties of PLA-KL and PLA-TANN composites.

Nanoindentation was applied to investigate the effect of KL and TANN on the surface mechanical properties of PLA-KL and PLA-TANN composites. Figure 7 shows the values of nanoindentation modulus and hardness as a function of concentration of added polyphenols. The values were analyzed in a depth range from 1000 to 1500 nm, where the substrate has no influence. The values at zero concentration correspond to the neat PLA material and are therefore the same for all added fillers. The addition of fillers increases the nanoindentation modulus and hardness by a maximum of 8.9% and 20.9%, respectively, but the effect of concentration is not significantly pronounced. The lowest concentration (0.5%) for both polyphenols gives similar values for modulus and hardness, with a further increase of filler content the values of the mechanical properties start to deviate. KL shows significantly lower values compared to TANN. Moreover, the maximum investigated concentration of additives leads to hardness values that are lower compared to the lower concentrations for both TANN and KL, but the modulus still slightly increases for TANN and decreases for KL. However, it can be seen that with increasing KL addition (from 0.5% to 2.5%) the modulus and hardness decrease slightly, which could be due to the distribution of the KL near the surface or in deeper layers of the film. This decrease correlates well with the decrease in the contact angle and the measurement of zeta potential at the maximum KL concentration.

Analysis of the values normalized to neat PLA material properties shows a higher effect of the TANN additive compared to KL.

Figure 8 demonstrates normalized values of both measured properties, nanoindentation modulus and hardness, compared to the properties of neat PLA. Normalization has been done by the modulus and hardness value of neat PLA for all materials. It is visible that the addition of polyphenols affects hardness more than modulus measured by nanoindentation. While at low concentration of 0.5% the effect of KL and TANN are the same, for higher concentration TANN causes more pronounced increase of both mechanical properties.

The differences between KL and TANN in PLA composites could be related to the size of filler particles, interfacial interaction, and crystallization. First, the hydrodynamic diameter of the filler particles measured by scanning electron microscopy (SEM) was measured to be about 600 nm for the particles of KL in the previous work [46], while the average size of the TANN particles was about 450 nm. Indeed, larger aggregates formed in the PLA-KL composites than in the PLA-TANN composites. The results also show that the dispersion and interfacial interaction between TANN and PLA are better than those between KL and PLA, leading to improved adhesion and final surface mechanical properties of the PLA-TANN films compared to the PLA-KL composite films. In general, incorporation of KL and TANN into PLA improved the final surface mechanical properties, while the effect is slightly less pronounced with KL and TANN loading.

### 3.2. Antioxidant Properties of the PLA-KL and PLA-TANN Composites

Natural antioxidant agents such as KL and TANN enhance the oxidation process and indeed improve the stability of thermoplastic composites. They reduce the concentration of reactive oxygen species and free radicals. Accordingly, antioxidant composites material accelerates the important preventing process in packaging as well as in biomedical applications. 

Figure 9 and Figure 10 show the dependence of UV-vis spectra of PLA-KL and TANN composites immersed directly in DPPH•-solution and measured at 517 nm, evaluated after 24 h (1, 12, 24 h). The results of antioxidant activity of PLA-TANN composites are shown in Figure 9. It can be clearly seen that neat PLA did not exhibit any antioxidant activity. The absorbance of neat PLA after 24 h slightly decreases and therefore provides around 17% of inhibition. With the incorporation of TANN into PLA, the antioxidant activity of the PLA-TANN composites increased and reached the max value after 24 h. Small differences between the PLA-TANN composites were measured, but the antioxidant activity is approximately independent of TANN loading. In general, the PLA-0.5 TANN composites after 24 h reach 91% of AO inhibition, the same value of AO inhibition 94% is measured for PLA-1.0 TANN composites. Furthermore, Figure 9 also shows that PLA-TANN composites containing 0.5 wt.% of TANN could reduce 50% of DPPH concentration after 10 h. Moreover, PLA-1.0 TANN and PLA-2.5 TANN composite films showed almost the same antioxidant activity up to about 17.5 h. The time-dependence study of the absorbance of PLA-2.5TANN composites measured every 3 min is also represented in Figure 9a.

Not only TANN but also KL is recognized as an efficient antioxidant accelerator in the polymer matrix. Figure 10 clearly shows the difference between the antioxidant activity of the PLA-KL composites with 0.5, 1.0., and 2.5% KL. As expected, the results represented that higher concentration of KL yielded PLA-KL composites with higher antioxidant activity during 12 h. With an increasing incorporation of KL in PLA, the antioxidant activity of the PLA-KL composites was enhanced and after 24 h only the PLA-KL composites with 2.5 wt.% reached the maximum value of inhibition. The other two PLA-KL composites with 0.5 and 1.0% of added KL provide 70% antioxidant activity after 24 h, with no significant differences between the composite activity.

The results of the antioxidant activities of PLA-TANN and PLA-KL composites shown in Figure 9 and Figure 10 followed with the same behavior of KL and TANN as an antioxidant accelerator, but with some differences; the PLA-TANN composites containing 0.5 wt.% of TANN were able to reduce 50% DPPH concentration after 10 h and on the other side the PLA-KL composites with 0.5 wt.% of KL after 15 h.

In all samples from PLA-TANN and PLA-KL, antioxidant activity is time-dependent and increases in the period from 1 h to 24 h. Only 0.5% of KL and TANN resulted in a halving of DPPH concentration, while the plateau is observed only in the samples with a higher value of KL and TANN; therefore, it is expected that the antioxidant activity of PLA-KL and PLA-TANN with 0.5 and 1.0% also increases when the 24 h is exceeded. The results indicate that the oxidative capacity of PLA-TANN and PLA-KL composites increases with increasing content of KL and TANN. Although our antioxidant evaluation of PLA-KL and PLA-TANN composites measured by DPPH assay showed the superior antioxidant properties of PLA-KL and PLA-TANN films, especially at a high loading of 2.5% of KL and TANN, the PLA-TANN composites enhanced the antioxidant activity more than the PLA-KL composites at low levels of added polyphenols. 

These results are consistent with a previous work that investigated the antioxidant properties of PLA-polyphenol composites. This showed that the antioxidant properties were improved as they were more efficient in reducing DPPH over time and increased with increasing polyphenol content compared with the neat PLA polymer [50,60,61,62,63,64,65,66,67]. However, our research provides some crucial differences between the measured antioxidant activities of PLA-KL and PLA-TANN composites that could be related to the properties explained below. In general, the antioxidant properties of polyphenolic fillers depend on the structural properties of KL and TANN (Mw, functional groups, polydispersity, and purity). The aliphatic hydroxyl groups in the side chains, non-esterified hydroxyl groups, polydispersity, and low purity decrease the radical scavenging activity of KL. Additionally, KL with less aliphatic hydroxyl groups and more phenolic hydroxyl groups, lower and narrower *Mw* distribution increased the antioxidant activity. Another reason for the altered antioxidant properties of KL could also be the length of the alkyl side chain in the phenylpropane units and in the carboxyl and alcohol groups, which increased the antioxidant activity [27,50]. 

In conclusion, in our study, the lower antioxidant activity of PLA-KL compared to PLA-TANN is probably related to the changes in polydispersity, low purity, and the presence of fewer phenolic groups compared with TANN, which accelerate the antioxidant behavior already measured in the study reported here [45,46]. The differences between the KL and TANN polydispersity were found: 0.64 for KL and 0.44 for TANN. Indeed, the bigger particles for KL (612 nm) in comparison with 454 nm of TANN contribute to more efficient surface available OH and COOH group with PLA that enhanced the interaction with polymer (measured in the study [45]). Moreover, the low purity of KL containing many residual carbohydrates that can generate hydrogen bonds with phenolic hydroxy groups of lignin and, therefore, interferes with the antioxidant activities of KL as was also studied in the literature [27,31,32,60,62,64,68,69].

### 3.3. Antibacterial Behavior of PLA, PLA-KL, and PLA-TANN Composites

The last aspect of multifunctional properties refers to the antibacterial properties of PLA-based KL and TANN composites. Regarding the charge of the polymer surfaces, typically positive charges in the polymers were found to be precursors for the antimicrobial activity and roughness of plastic surfaces altering the surface properties.

The antimicrobial assay of the neat PLA as a reference sample and the PLA composite samples containing KL and TANN filler (in the three different concentrations 0.5%, 1.0%, and 2.5%) was investigated against a model pathogen, namely *E. coli*, after 1 h, 6 h, and 24 h and is shown in Table 3. Accordingly, it can be seen that the antibacterial activity of the PLA composites in the presence of KL and TANN was much more evident than that of the neat PLA control, which showed resistance to the adhesion of *E. coli*, while neat PLA showed no antimicrobial activity, as expected. 

The antibacterial activity of the PLA-TANN (Table 3) composites monitored after 24 h showed that only the composite with 2.5% TANN achieved an antimicrobial activity of 88.5%. The composites with lower TANN content in PLA, i.e., 1.0% and 0.5%, achieved lower values of bacterial activities, namely 58% and 20%, respectively. It seems that the efficiency of antimicrobial activity of PLA-TANN composites increases with increasing filler content. In contrast, composites with KL, which were incorporated into the PLA composites, did not show the same trend against *E. coli* as PLA-TANN. The results of the antimicrobial activities of PLA-KL with a high content of added KL (2.5%) exhibited a lower antibacterial activity than neat PLA and thus showed no antimicrobial activities, but the PLA-KL composites with only 0.5% showed an antibacterial activity of almost 82% after 24 h.

The results also represent that the composites containing TANN and KL exhibited bacterial resistance to *E. coli* regardless of filler loading. A clear dependence on antibacterial activity is not evident. On the other hand, the antibacterial activity of PLA-KL is related to its origin, the polyphenol content, the presence of the phenolic compound and, in particular, the different functional groups containing carbonyl and hydroxyl groups considering the published research [70]. In our study, the effect of the incorporation of KL and TANN into PLA not only depends on the polyphenolic content and the carbonyl and hydroxyl groups, but the effect was also attenuated by the roughness of the PLA-KL and TANN composites and the presence of an active polyphenolic filler near the PLA surface. As noted in the study of metal-based nanoadditives into the PLA matrix [37], the presence of an active metal oxide filler on the surface of PLA altered the antibacterial properties of PLA composites against *E. coli*. Moreover, the rough surfaces with larger surface area inhibited the bacteria more.

Our results also indicate that bioactivity in the form of antioxidant and antimicrobial properties is not diminished by surface parameters. It has been shown that most of the monitored surface properties change more significantly when the amount of added TANN and KL is decreased. Conversely, antioxidant activity is higher when the added amount of polyphenols is higher, while for antimicrobial activity neither the added mass of polyphenols nor a change in surface parameters has a clear influence.

## 4. Conclusions

In the present work, the preparation of sustainable and renewable PLA-based KL and TANN composites by hot melt extrusion was investigated. The synergistic effect of the incorporation of KL and TANN on the final surface mechanical properties as well as antioxidant and antimicrobial activity of these composites was monitored. Moreover, the incorporation of KL and TANN into PLA with three different contents proved to be an effective method to enhance the surface mechanical properties of neat PLA, resulting in a larger surface effect depending on the polyphenol loading, the size of the particles with available functional carbonyl and carboxyl groups on the surface, and the interaction between the polyphenol filler and PLA.

The results showed that the increased contact angle, roughness, and zeta potential of the PLA-based KL and TANN could be related to the differences in the distribution of polyphenols near the surface of the PLA composites. In addition, all PLA composites showed hydrophilic character, which was more pronounced in PLA-TANN than in PLA-KL. The zeta potential of the PLA-KL and PLA-TANN composites exhibited an anionic charge throughout the pH range and correlated well with the hydrophilic nature of the PLA-KL and TANN composites. The highest surface concentration of polyphenolic filler resulting in a higher surface charge at the lowest content compared with neat PLA was evaluated, confirming the preferential distribution of polyphenolic filler near the surface, and decreased with filler loading as well as in the depth of the PLA polymer composites themselves. Moreover, the nanoindentation measurements confirmed the performance of both nanoindentation modulus and hardness for PLA-KL and PLA-TANN by a maximum of 8.9% and 20.9%, respectively, with a higher effect of the TANN additive compared to KL.

Finally, the synergistic and multifunctional effect of PLA-KL and PLA-TANN composites was also followed/supported by antioxidant and antibacterial activity. In addition, the incorporation of KL and TANN into PLA resulted in the migration of radical scavengers, which was monitored by DPPH assay; the kinetic study showed increased antioxidant activity, which increased with polyphenol loading. The integration of KL and TANN into PLA was sufficient for the preparation of antimicrobially active PLA-based KL and TANN composites against *E. coli*, but the dependence of the polyphenol filler content on the final antibacterial properties has not yet been clearly elucidated.

## Figures and Tables

**Figure 1 polymers-14-01532-f001:**
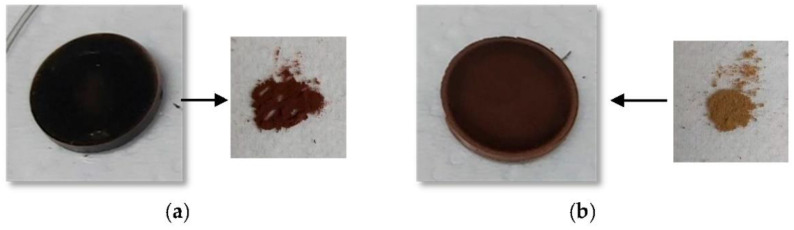
Images of the pellet sample of (**a**) Kraft lignin and (**b**) tannin.

**Figure 2 polymers-14-01532-f002:**
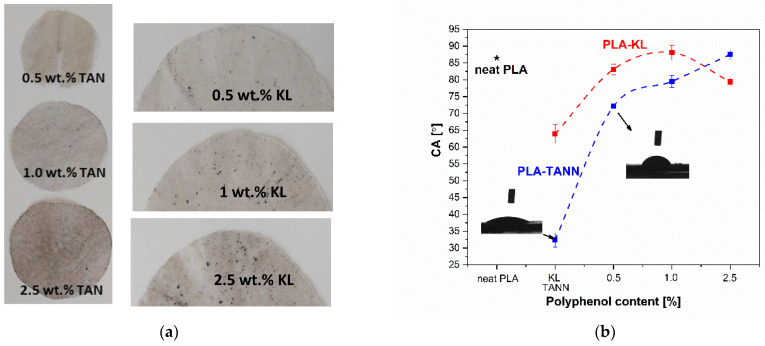
Images samples of (**a**) PLA-TANN and PLA-KL composite films using different KL and TANN concentrations. The wettability of (**b**) neat PLA, KL, TANN, PLA-KL, and PLA-TANN composites as a function of wt. % of KL and TANN loading. Star marker represents neat PLA material.

**Figure 3 polymers-14-01532-f003:**
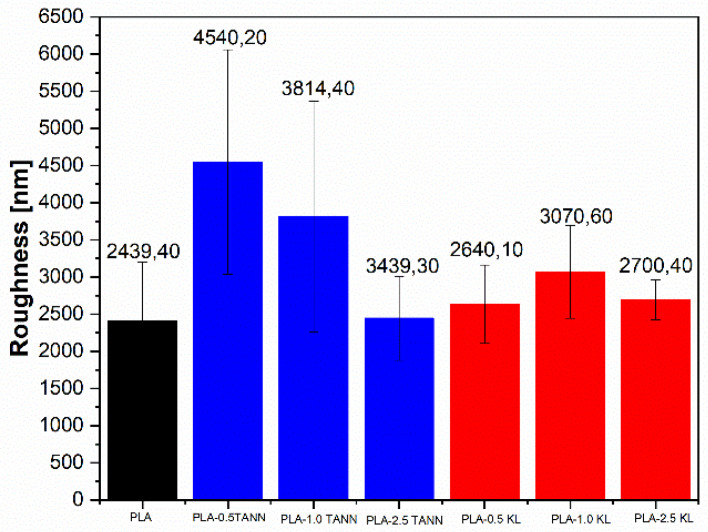
An average roughness value of neat PLA, PLA-TANN, and PLA-KL composites.

**Figure 4 polymers-14-01532-f004:**
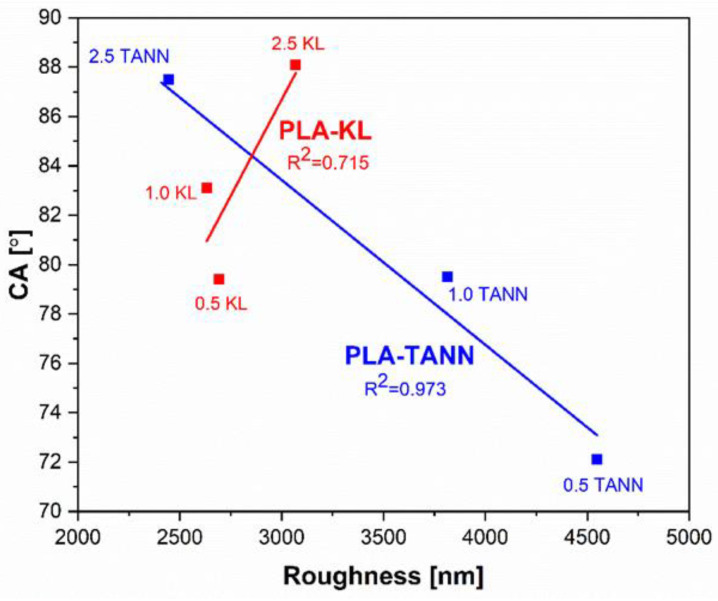
Correlation between CA and roughness of PLA-KL and PLA-TANN composites.

**Figure 5 polymers-14-01532-f005:**
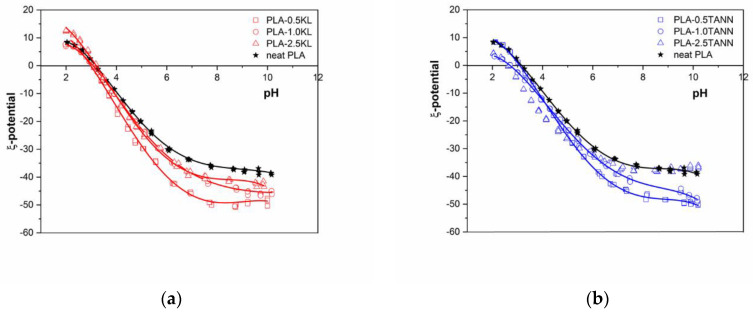
Zeta potential of (**a**) PLA-KL composites and (**b**) PLA-TANN composites in the range of pH 2–12.

**Figure 6 polymers-14-01532-f006:**
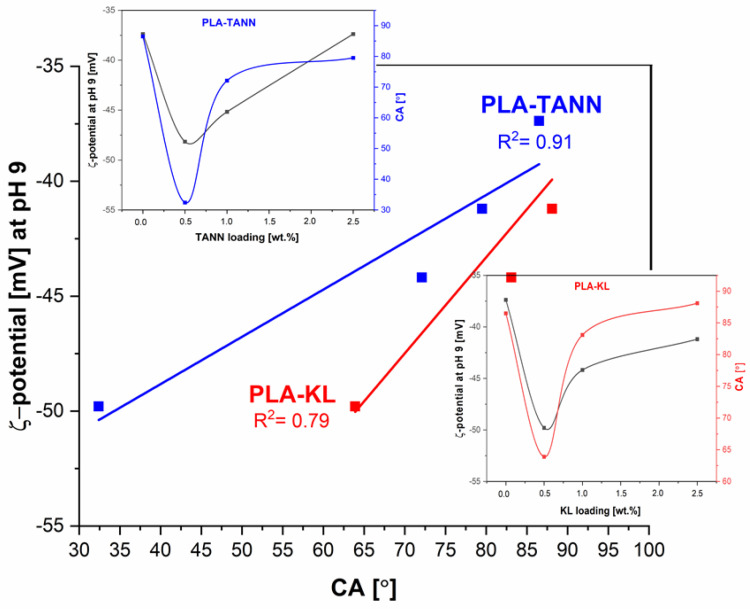
The correlation between the plateau zeta potential at pH 9 and polyphenolic filler loading in PLA-KL and PLA-TANN composites. The insets show the dependence on CA, zeta potential at pH 9, and polyphenol loading.

**Figure 7 polymers-14-01532-f007:**
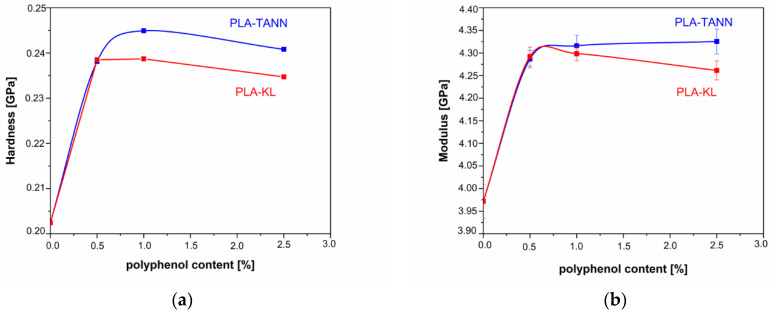
Nanoindentation hardness (**a**) and modulus (**b**) in respect to concentration of added polyphenols.

**Figure 8 polymers-14-01532-f008:**
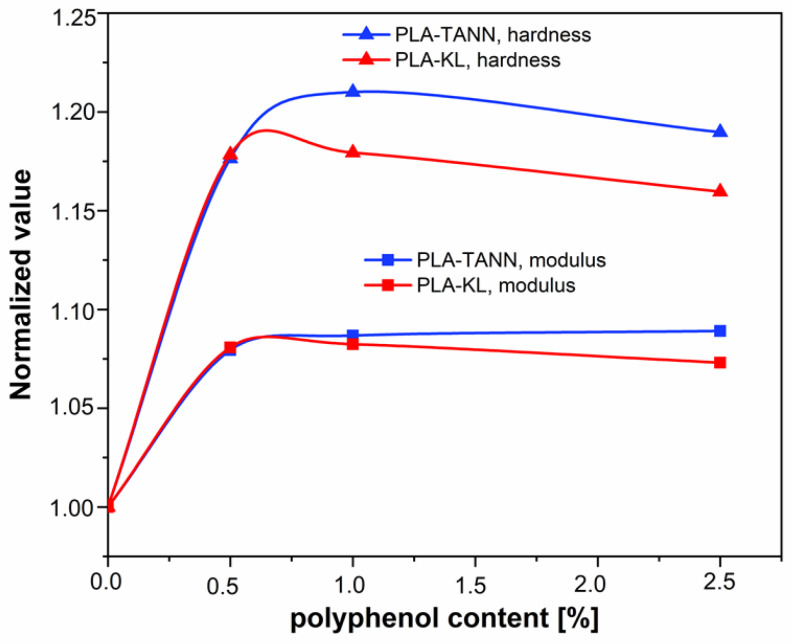
Normalized properties of neat material values of nanoindentation modulus and hardness are plotted in dependence on concentration of added polyphenols.

**Figure 9 polymers-14-01532-f009:**
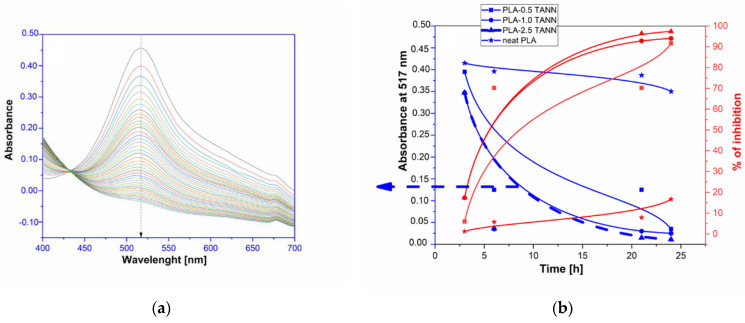
(**a**) The absorbance of PLA-2.5 TANN composites as a function of Wavelength (blue) and (**b**) reaction kinetics and % of inhibition of the PLA based TANN composites film evaluated with DPPH radical scavenging in the methanol solution indicated during 24 h (red).

**Figure 10 polymers-14-01532-f010:**
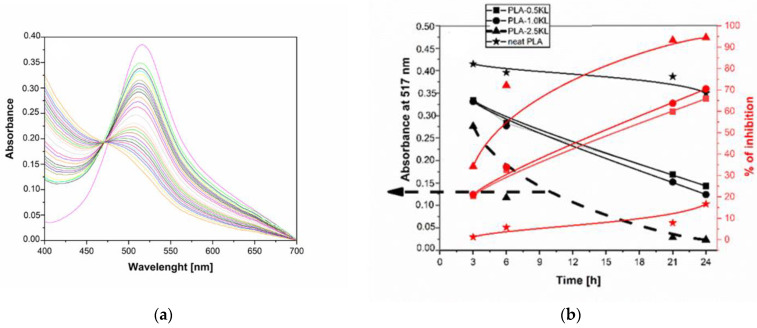
(**a**) The absorbance of PLA-KL composites as a function of Wavelength and time, (**b**) reaction kinetics of the PLA based KL composites film evaluated with DPPH radical scavenging in the methanol solution indicated during 24 h.

**Table 1 polymers-14-01532-t001:** Prepared PLA-KL and PLA-TANN composites.

Material Description/Polyphenolic Filler Loading [wt. %]	0.5	1.0	2.5
PLA-based Kraft lignin	PLA-0.5KL	PLA-1.0KL	PLA-2.5KL
PLA-based tannin	PLA-0.5TANN	PLA-1.0TANN	PLA-2.5TANN

**Table 2 polymers-14-01532-t002:** Testing parameters of Nanoindenter G200.

Tip Geometry	Depth [nm]	Distance between Indents [µm]	Poisson’s Ratio	Number of Indents
Berkovich	2500	100	0.35	36

**Table 3 polymers-14-01532-t003:** Antibacterial assay of Neat PLA, PLA-KL, and PLA-TANN composites.

Composites Name/% of Inhibition in Time	t_1_ [1 h]	t_2_ [12 h]	t_3_ [24 h]
neat PLA	−14.11	−14.11	−14.11
PLA-0.5 TANN	1.78	20.87	20.87
PLA-1.0 TANN	13.02	58.1	58.1
PLA-2.5 TANN	14.2	88.11	87.21
PLA-0.5 KL	7.4	67.86	82.96
PLA-1.0 KL	3.25	54.27	36.52
PLA-2.5 KL	6.07	−4.85	−145.82

## Data Availability

Not applicable.

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
