# Peer review of "Kraft Lignin/Tannin as a Potential Accelerator of Antioxidant and Antibacterial Properties in an Active Thermoplastic Polyester-Based Multifunctional Material"

_polymers, 2022, doi:10.3390/polym14081532_

Round 1
Reviewer 1 Report
The manuscript is focused on development of poly (lactic acid) composites based on kraft lignin and tannin as natural fillers, manufactured by the hot-melt extrusion method, and investigation of their physical, chemical and mechanical properties, as well as their antioxidant and antimicrobial activity. In general, the manuscript is well-written, structured and informative, but still needs some minor improvements before acceptance for publication in the Polymers Journal. Please, see below my comments on your work:
In general, the title (lines 2-4), abstract (lines 20 to 36) and the keywords (lines 37-38) correspond to the aims and objectives of the manuscript.
The abstract is concise, specific, and outlines the aim and main results of the research.
Line 69-71: please justify why did you select kraft lignin from all other technical lignin types. i.e. organosolv lignin, hydrolysed lignin, lignosulfonates, soda lignin?
Please add some information on the Kraft process, chemical and physical properties of Kraft lignin.
Lines 95-96: please add the full terms of the abbreviations used - PBS and PCl. Poly(butylene succinate) and Poly(caprolactone)?
Line 114: Is there any reason for writing “antioxidant activity” in Bold and Italic?
Line 118: Mw – molecular weight?
In general, the Introduction part provides relevant information on the research topic.
Line 173: please explain why did you select the natural filler loading percentages, i.e. 0.5%, 1.0% and 2.5%? Is it based on literature data or previous findings of your experimental work?
Line 191: please provide data on the goniometer equipment used (company producer, city, country).
Line 205: please revise the caption of Figure 1. Now there is one unnecessary “a)”.
Line 229: Table 2 should be written in capital letters, please revise.
Line 261: Please add the standard used, i.e. ISO22196, in the references.
Overall, the Materials and Methods section is very well presented, detailed and informative.
Line 272: Please revise the title of section 3 to Results and Discussion.
Line 317: please add the full term, i.e. scanning electron microscopy, followed by the common abbreviation SEM.
Line 324: Figure 2 a is too small and nothing can be seen. Please replace it with a larger and better quality image.
In general, the Results section is quite well developed and discussed with relevant research works in the field.
Overall, the Conclusion part (lines 563-594) reflects the main findings of the manuscript.
The References cited are appropriate to the research topic. The references are neither properly cited in the main text of the manuscript, nor properly formatted in accordance with the journal requirements, please check the Instructions for authors.
Best regards!
Reviewer 2 Report
Authors fabricated PLA based composite with kraft lignin and tannin using hot-melt extrusion. This investigation is valuable. Before publishing the next level many things need to correct in the right way (Major revision). English needs to be polished throughout the manuscript.
Comments
The title of each word's first character should be capitalized as per journal guidelines. Please double-check the guidelines. Correspondence email id should be written properly, there are 2 authors corresponding authors please indicate which authors have which email.
References format is wrong. No “Title”, no bold all, and others error, check carefully.
Line 25 the anti- 24 oxidant/antimicrobial activity why / it should be and “activities”.
A background sentence should be included in the abstract first sentence.
abstract should be rewritten in a quantitative manner.
The reference style is the wrong Ex: Line 47 it should be [1-6]. Check another place inside the manuscript and correct it accordingly.
The introduction section is too long. Make it short and less paragraph maximum 4. Novelty should be highlighted in the introduction section. This suggested recent references are valuable to include in the introduction section based on PLA https://doi.org/10.1016/j.indcrop.2020.112320, https://doi.org/10.1155/2021/4933450, https://doi.org/10.1016/j.matdes.2020.108603
The last paragraph should be describing what the authors did in this manuscript.
There are so many references authors included in this manuscript please make it a maximum of 50-60.
Table 1 is complicated to understand authors should have revised it as sample code, description, and % of raw materials like this.
Line 184, the section should be 2.2.
Line 239, insert recent references International journal of Biological macromolecules 136 (2019) 661-667 Including 73 references number.
In this manuscript equation is only one so no need for numbering. Please make a (×) mark inside the equation (Before 100%).
Section 2.2.1.3 procedure for the antibacterial activity should be rewritten clearly. There are many references inside. Make it into one.
Line 268, Polyphenolic, p should be small.
Section 3, should be results and discussion.
Line 283-286, English is not correct.
Fig. 2 clarity is so poor and small, not visible. Make it a better way.
Line 321 has so many references. Remove it.
The abbreviation of Contact angle author wants to use as CA, but not defined in the manuscript. Please define it first then use its shot form.
Fig 3 need to be statically analyzed by ANOVA.
How authors did normalize? figure 8.
Line 462, hours should have written as h. Check another place.
Authors need to discuss deeply including showing the mechanism why TANN/PLA shows high antioxidant activity than other composites. In section 3.2 cite it Cellulose, 29 (2022) 2399-2411), this suggested reference well described how antioxidant value depends on varying time and dose.
What the mechanism for antibacterial study authors needs to described in the manuscript. Why TANN-based composite shows higher antibacterial activity. Authors need to described reason on compare basis.
How about the antibacterial activity of gram-positive bacteria.
The conclusion section needs to be short and precise.
Reviewer 3 Report
Recommendation: Minor revisions needed.
Comments:
The paper by Črešnar et al. contributes the aspect of biodegradable composite material based on PLA, PLA-KL and PLA-TANN reinforced with different ratios of natural filler loadings. The title and abstract are appropriate for the content of the text. Moderate English changes are required. The article gives an interesting historical and scientific perspective on composite materials.
Some issues should be addressed before publication.
- Table 1. This table is not necessary. You can describe the terminology/abbreviation in a few sentences.
- Figure 1. Please label image a/b on your graph. Please rephrase your title.
- Page 5, Line 226. “Continuous Stiffness Measurement is a nanoindentation characterization method developed by Oliver and Pharr, which utilizes the dynamic loading of the sample material.” Bold font is not necessary here.
- Figure 2. Please change the font size in image a.
- Figure 3. Please add the roughness data on the top of each column.
Round 2
Reviewer 2 Report
The authors improved the manuscript. Still there are problems that need to sort out before final publications.
- In the title last word material; ‘m’ character should be capital.
- References style still wrong. Here reviewer provided, please check carefully again ‘Polymers’ journal authors guidelines. Authors need to follow strictly journal guidelines.
References
References must be numbered in order of appearance in the text (including citations in tables and legends) and listed individually at the end of the manuscript. We recommend preparing the references with a bibliography software package, such as EndNote, ReferenceManager or Zotero to avoid typing mistakes and duplicated references. Include the digital object identifier (DOI) for all references where available.
Citations and references in the Supplementary Materials are permitted provided that they also appear in the reference list here.
In the text, reference numbers should be placed in square brackets [ ] and placed before the punctuation; for example [1], [1–3] or [1,3]. For embedded citations in the text with pagination, use both parentheses and brackets to indicate the reference number and page numbers; for example [5] (p. 10), or [6] (pp. 101–105).
- Author 1, A.B.; Author 2, C.D. Title of the article. Abbreviated Journal Name Year, Volume, page range.
- Author 1, A.; Author 2, B. Title of the chapter. In Book Title, 2nd ed.; Editor 1, A., Editor 2, B., Eds.; Publisher: Publisher Location, Country, 2007; Volume 3, pp. 154–196.
- Author 1, A.; Author 2, B. Book Title, 3rd ed.; Publisher: Publisher Location, Country, 2008; pp. 154–196.
- Author 1, A.B.; Author 2, C. Title of Unpublished Work. Abbreviated Journal Name year, phrase indicating stage of publication (submitted; accepted; in press).
- Author 1, A.B. (University, City, State, Country); Author 2, C. (Institute, City, State, Country). Personal communication, 2012.
- Author 1, A.B.; Author 2, C.D.; Author 3, E.F. Title of Presentation. In Proceedings of the Name of the Conference, Location of Conference, Country, Date of Conference (Day Month Year).
- Author 1, A.B. Title of Thesis. Level of Thesis, Degree-Granting University, Location of University, Date of Completion.
- Title of Site. Available online: URL (accessed on Day Month Year).
- There are many references in his manuscript now it is 96 which is unacceptable. References wording 1/3 of the total words of the manuscript. Many places have unnecessary references.
- The authors have not inserted this reference but it is mentioned that it is inserted which is ridiculous. Please check again. “Line 239, insert recent references International journal of Biological macromolecules 136 (2019) 661-667 Including 73 references number. The reference was revised”.
- In this manuscript equation is only one so no need for numbering. Please make a (×) mark inside the equation (Before 100%). The suggested numbering of the equation was changed to a. I told to put a crossmark (×). It does not mean numbering the equation as (a)
What is the meaning of (a), need to remove it. Need to revise it, as line 225 “The radical scavenging activity was calculated as per the following expression’’.
